# A multi-criteria decision-making framework for managing the safety of marine recreational powered platforms: Integration with the SHELL model

Shao-Hua Hsu[1]*, Yo-Kang Yang[2], Ya-Fan Ho[1], Meng-Tsung Lee[2], Jao-Chuan Lin[2]*

1 Department of Marine Environment and Engineering, National Sun Yat-sen University, Kaohsiung, Taiwan, 2 Department of Marine Leisure Management, National Kaohsiung University of Science and Technology, Kaohsiung, Taiwan

* kylehsu5678@gmail.tw (S-HH); jcl@nkust.edu.tw (J-CL)

## Abstract

The rise of marine recreational activities has led to a growing use of marine recreational powered platforms, raising safety concerns related to navigation. In Taiwan, the current regulatory system for such platforms remains fragmented and under debate. This study aims to support policy development by identifying key safety management priorities. This study utilized the four core components of the SHELL model, which include Software, Hardware, Environment, and Liveware, as the analytical foundation and identified 20 preliminary safety criteria through an extensive review of relevant literature. A Modified Delphi Method and DEMATEL analysis were applied to gather expert insights and prioritize 10 representative indicators. The resulting Influence Network Relation Map revealed that "Comprehensive Management Regulations" had the highest causal influence across all dimensions. Additionally, "Basic Navigation Concepts" and "Emergency Response and Safety Knowledge" were found to be the most central elements. Based on these findings, the study recommends targeted measures including enhanced regulation, improved training, radar monitoring, and spatial planning to reduce navigation risks and promote safer marine recreation. Building on the above findings, this study confirms the effectiveness of an innovative integration of the SHELL model and the DEMATEL method, which provides a structured and adaptive framework capable of systematically identifying systemic navigational risks in marine recreational activities.

## Introduction

In recent years, with rapid global economic growth, elevated living standards, and increased demand for leisure tourism, there has been a continuous rise in population engaging with coastal national scenic areas, swimming beaches, tourist fishing

**Data availability statement:** All relevant data are within the manuscript and its Supporting Information files.

**Funding:** The author(s) received no specific funding for this work.

**Competing interests:** The authors have declared that no competing interests exist.

ports, and maritime recreational activities [1–3]. Consequently, following the trend of increased maritime activities, the demand for various powered platforms has correspondingly increased, diversifying marine leisure activities and enhancing maritime recreational experiences [4]. These include PORTA-BOTE, Quickboats, Insta-Boat, Uui-Float fishing vessels, Go-kart boats, Kayaks, Inflatable boats, and Styrofoam vessels [5]. These platforms are characterized by their portability, economic accessibility, and ease of acquisition. However, the popularization of such powered platforms has led to the emergence of associated safety concerns and potential risks. This is particularly critical in navigation safety incidents, which invariably involve the lives and property of vessel occupants. Such incidents may result not only in vessel damage and submersion but also in personnel injury, maritime disappearances, and fatalities [6–9].

In Taiwan's maritime waters from 2016 to 2023, there were 1,046 reported casualties during water recreational activities, with an additional 160 powered platform rescue operations conducted. With an average of approximately 20 powered platform incidents annually, these equipments are increasingly recognized as potential navigational hazards [10,11]. Consequently, in Taiwan's maritime zones, which encompass critically important shipping channels, powered platforms present a significant challenge to navigational safety. Unlike crew members of large commercial vessels, yachts, or cruise ships who must undergo formal training in accordance with the 《International Convention on Standards of Training, Certification and Watchkeeping for Seafarers》, operators of these platforms often lack adequate navigational safety awareness. This deficiency underscores the imperative necessity for conducting comprehensive research into the navigational safety risks associated with powered platforms.

According to Taiwan's current "Water Recreation Activity Management Regulations" and "Safety Guidelines for Powered Platforms and Operator Requirements," water recreational activities are categorized into three classifications based on their characteristics: "Activities Requiring Powered Platforms," "Activities Not Requiring Powered Platforms," and "Shore Fishing Activities." The activities requiring powered platforms are further subdivided as follows [12]:

- Activities utilizing human-powered platforms: kayaking, stand-up paddling.

- Activities utilizing natural-powered platforms: surfing, windsurfing, kitesurfing.

- Activities utilizing mechanically-powered platforms: water skiing, parasailing, banana boats, towable tubes, personal watercraft, inflatable boats, and other powered platforms.

Furthermore, distinct regulatory requirements are established for powered platforms based on passenger capacity: "2 or fewer persons" versus "3 or more persons." Operators of powered platforms with a capacity of 3 or more persons are required to possess a powered small vessel or yacht operator's license. Conversely, operators of powered platforms with a capacity of 2 or fewer persons must obtain educational training certification from government-accredited professional institutions prior to

operation. However, despite these regulatory provisions established by governmental authorities, significant regulatory gaps persist. Notably, when individuals engage in water-related activities using self-manufactured powered platforms or independently purchased powered platforms, insufficient regulatory oversight may result in latent safety risks, contributing to frequent accident occurrences.

The SHELL model provides a framework for examining the interconnections between personnel, equipment, and environment in high-risk contexts. The model's nomenclature derives from the initial letters of its dimensional components: Software, Hardware, Environment, Liveware and Liveware, enabling identification of interdimensional relationships and their impact on system performance and safety [13]. This framework has been extensively applied in analyzing human factors in aviation safety [14–17], medical occupational risk factors [18–20], and other risk management methodologies [21,22]. In maritime incident research, studies have primarily focused on human factors in vessel accidents [23–25], with limited investigation into navigational safety aspects of water recreational platforms. Therefore, this study advocates the application of the SHELL model to assess the safety of powered platforms. Unlike the commonly employed HFACS-MA (Human Factors Analysis and Classification System–Maritime Adaptation) framework [26–28] in maritime safety research, the SHELL model offers a systems-oriented perspective that integrates four critical dimensions: personnel, equipment, environment, and interpersonal interaction. This integrative approach effectively addresses the limitations of existing risk assessments for recreational activities, which often tend to be overly technical or narrowly focused.

According to International Maritime Organization statistics, human factors directly or indirectly contribute to over 80% of maritime navigational incidents [29]. These incidents typically result not from singular causation but from complex concatenations of multiple factors [30,31]. The European Union's vessel safety assessment framework categorizes navigational risk factors into four dimensions: Personnel, Hardware, Software, and Environment [32], highlighting the necessity of conducting multidimensional and systematic risk analyses. For instance, the SHEL model has been applied in studies investigating human-related maritime incidents and in developing preventive strategies for ship collision avoidance [33–35].

This study employs the SHELL model as its theoretical framework to investigate the risks associated with marine recreational powered platforms. Through a systematic examination of the internal dynamics among the four key elements—Software, Hardware, Environment, and Liveware—the study conducts a detailed analysis of their interrelationships in order to establish effective, evidence-based decision-making protocols for safety management.

Given that this research specifically focuses on recreational powered platform activities characterized by non-professional users and a lack of formal regulatory oversight, particular emphasis is placed on the interactions between Liveware–Liveware and Environment–Liveware dimensions. This analytical orientation marks a critical departure from conventional studies that predominantly emphasize technical or operational aspects, enabling a more accurate reflection of the risk patterns inherent in informal maritime practices especially those shaped by experiential disparities among users and environmental uncertainties.

Grounded in the overarching goal of enhancing maritime safety, this study seeks to develop a comprehensive safety management criteria framework tailored to the operational realities of powered platform usage in water-based recreational activities. By identifying and analyzing the correlations among key risk management indicators, the research aims to formulate an integrated strategy framework that can support more responsive and informed decision-making processes for marine recreation management authorities.

## Methods

With the advancement of various industries, increasing attention has been given to the development and application of safety management evaluation frameworks across sectors to enhance responses to critical types of risk. In particular, research has emphasized the importance of systematically integrating occupational and process risk assessments [36]. To address uncertainties inherent in such evaluations, researchers have adopted a wide array of quantitative and managerial techniques, including credibility-weighted expert judgment, the Delphi method, fuzzy logic, Bayesian inference, sensitivity

analysis, and fuzzy number scoring [36–40]. Despite these advancements, uncertainty remains a significant challenge in safety management due to factors such as discrepancies in expert judgment, data insufficiencies, and semantic ambiguities in information.

To mitigate such uncertainty-related issues, this study adopts the SHELL model, Delphi method, and DEMATEL approach. The SHELL theoretical framework ensures that the proposed safety management criteria comprehensively incorporate the dimensions of personnel, hardware, environment, and software, thus providing a rational structure for safety assessment. The Delphi method facilitates the integration of expert experience, professional insight, and collective intelligence, reducing individual bias and enabling the transformation of high-uncertainty scenarios into stable and high-consensus safety judgments [40]. Meanwhile, the DEMATEL method is applied to construct a causal network among risk criteria, clarifying the interrelationships among safety management indicators and assisting in the identification of key influence pathways.

This study adopted social science research methodologies for conceptualization, analysis, comparison, and assessment, supplemented by expert interviews, to synthesize and consolidate findings, ultimately proposing concrete and feasible recommendations. This study conducted a two-phase questionnaire survey. The first phase was administered on September 29, 2024, and concluded on October 11, 2024. The second phase was carried out on October 15, 2024, and completed on October 27 of the same year.

## Modified delphi method (MDM)

The MDM optimizes the traditional Delphi process by streamlining the investigation procedure. MDM eliminates complex investigation iterations by utilizing relevant literature and expert interviews to design a semi-structured first-round survey, significantly enhancing expert response rates and reducing inconsistent investigation outcomes. This approach maintains the essential elements of the traditional Delphi method while streamlining its more complex processes, enabling scholars to focus more effectively on research objectives [41,42].

The implementation process of the MDM comprises five sequential steps:

**Step 1**: Consolidate investigation content through literature review and design the survey using rating scales.

**Step 2**: Identify and establish a Delphi expert panel comprising scholars well-versed in the research topic and elucidate the research objectives.

**Step 3**: Establish implementation criteria for the MDM, distribute and collect investigation.

**Step 4**: Synthesize collective expert panel opinions through quantitative analysis, and request panel members to provide feedback or modifications.

**Step 5**: Verify consensus consistency in investigation results. If consensus is achieved, conclude the investigation phase; if consensus is not reached, adjust and modify investigation content and repeat steps 3 and 4 until consensus is attained.

Consequently, to ensure smooth investigation administration, this study employed the MDM for investigation design and survey implementation, utilizing Likert-Type scale for expert panel responses. The scale points correspond to: "1 - Very Unimportant," "2 - Unimportant," "3 - Neutral," "4 - Important," and "5 - Very Important," where higher scores indicate stronger agreement. The mean value for each criteria factor represents its average intensity.

For evaluating expert group consensus levels regarding criteria factors, this study employed the Coefficient of Variance (CV) as the expert judgment criterion. Criteria factors with mean values exceeding 4 and coefficient of variance less than or equal to 0.5 were designated as representative risk management criteria for this study. Furthermore, to verify consensus achievement among all expert panel members, the Consensus Deviation Index (CDI) was calculated for investigation criteria content [43,44].

- Assuming that in the *t*th round of the MDM survey, the score given by the *h*th expert for the *j*th item is $X^{jht}$, then the mean and standard deviation of *R* experts' scores for *j* items in the *t*th round investigation, denoted as $\overline{X^{jt}}$ and $S^{jt}$ respectively, are expressed as follows:

$$\overline{X^{jt}} = \frac{1}{R} \sum_{h=1}^{R} X^{jht}, \forall j, t \tag{1}$$

$$S_{jt} = \sqrt{\frac{1}{(R-1)} \sum_{h=1}^{R} \left( X^{jht} - \overline{X}^{jht} \right)^2}, \forall j, t \tag{2}$$

- For investigation round *t*th, the coefficient of variation $CV^{jt}$ for item *j* is calculated as follows:

$$CV^{jt} = \frac{S^{jt}}{\overline{X^{jt}}}, \forall j, t \tag{3}$$

- The consistency deviation index is calculated as follows, where a smaller numerical result indicates higher consensus among expert panel members regarding the specific criteria factor.

$$CDI^{jt} = \frac{S^{jt}}{max_j \left( \overline{X^{jt}} \right)}, \forall j, t \tag{4}$$

## DEMATEL

The DEMATEL methodology originated in 1972 at the Battelle Memorial Institute of Geneva, Switzerland, effectively integrates expert knowledge and utilizes the Influential Network Relation Map (INRM) to elucidate the causal relationships and influence intensity among various criteria. Through visual representation, it assists decision-makers in analyzing, identifying, and formulating optimal management decisions and improvement strategies [45,46].

Therefore, based on DEMATEL's methodology for analyzing the characteristics of issues and their criteria, as well as the interrelationship intensity among these criteria, the implementation procedures are presented as follows:

### Step 1: Relational Factors Between Criteria and Establish Measurement Scale:

This research initially screened preliminary criteria through Rough Set Theory, followed by the MDM to establish and clearly define the criteria. Subsequently, through expert investigation, the inter-relational influence values between criteria were obtained. A five-point measurement scale of 1, 2, 3, 4, and 5 was employed, representing "Very low influence (1)", "Low influence (2)", "Moderate influence (3)", "High influence (4)", and "Very high influence (5)", respectively [34].

### Step 2: Construct Direct-Relation Matrix:

The Direct-Relation Matrix is derived from investigation responses. When there are *n* criteria, the degree of influence between each pair of criteria is compared, resulting in an n × n $n \times n$ Direct-Relation Matrix X, where $X_{ij}$ represents the degree of influence from criteria *i* to criteria *j*. Since the diagonal elements of matrix X represent self-influence and are set to 0, the Direct-Relation Matrix X is expressed as:

$$X = \begin{bmatrix} 0 & X_{12} & \cdots & X_{1n} \\ X_{21} & 0 & \cdots & X_{2n} \\ \vdots & \vdots & \ddots & \vdots \\ X_{n1} & X_{n2} & \cdots & 0 \end{bmatrix}$$

(5)

**Step 3: Construct Normalized Direct-Influence Matrix):**

In this step, the standardized influence matrix is denoted as N, with the standardization criterion value set as $k$. The calculation of the standardized direct influence matrix N is as follows:

$$N = kX$$

(6)

$$k = \frac{1}{max\left[\max\limits_{1 \le i \le n} \sum_{j=1}^{n} X_{ij}, \max\limits_{1 \le j \le n} \sum_{i=1}^{n} X_{ij}\right]}$$

(7)

**Step 4: Calculation of Total-Influence Matrix:**

The Total-Influence Matrix T is calculated as follows, where I represents the identity matrix:

$$\lim_{s \to \infty}(N + N^2 + N^3 + \cdots + N^s) = N(I - N)^{-1}$$

(8)

**Step 5: Calculation of $d_i$ and $r_j$ values:**

Through matrix row and column operations, calculates the values of di and $r_j$ by summing each row and column respectively. Here, $d_i$ represents the row sum of the total influence matrix $T$, while $r_j$ represents the column sum. Let $T_{ij}$ denote the criteria factor in the total influence matrix T, where $i, j \in \{1, 2, 3,..., n\}$. The calculation of row and column sums in the total influence matrix $T$ can be expressed as follows:

$$d_i = \sum_{i=1}^{n} t_{ij} \qquad (i = 1,2,3,\cdots,n)$$

(9)

$$r_j = \sum_{j=1}^{n} t_{ij} \qquad (j = 1,2,3,\cdots,n)$$

(10)

**Step 6: Calculation of $(di+rj)$ and $(d_i-r_j)$ values:**

Based on the calculated values of di and $r_j$ the values of $(d_i + r_j)$ and $(d_i - r_j)$ can be obtained, where $(d_i + r_j)$ represents the prominence index, indicating the total degree to which an criteria influences and is influenced by other criteria. This enables the understanding of the criteria's central role in the issue under investigation. The $(d_i + r_j)$ value, known as the relation index, represents the net effect that distinguishes the degree to which criteria influences others versus being influenced by others. This relation index facilitates the classification of criteria into cause and effect groups. When the relation index is positive, the criteria is categorized as a cause criteria; conversely, a negative value designates the criteria as an effect criteria [47].

**Step 7: Construction and Analysis of Influence Network Relation Map:**

The Influence Network Relation Map is constructed with centrality $(d_i + r_j)$ as the horizontal axis and the net effect degree $(d_i − r_j)$ as the vertical axis. The coordinates are determined by the centrality value and net effect degree of each criteria. The map is divided into 4 quadrants by two perpendicular lines: one representing the mean centrality value and the other where the net effect degree equals 0. Through the distribution pattern of criteria across these quadrants, one can analyze the causal relationships, core significance, and inter-criteria relationships.

## Preliminary criteria framework

### MDM expert panel

The MDM represents the collective decision-making of expert groups. Given that this study explores a complex topic involving multiple dimensions, including human factors, powered equipment, and environmental conditions, it can be categorized as a research area with substantial variability in perspectives. As such, limiting the number of expert participants to between 5 and 10 individuals is considered appropriate [48,49].

This study employed a purposive sampling strategy to assemble a multidisciplinary panel of experts, with the aim of ensuring the validity of expert input and the relevance of insights to the specific context of marine recreational powered platform risk management. Expert selection was guided by clearly defined criteria, emphasizing substantial professional experience and academic specialization closely aligned with the study's focus. The panel was intentionally composed to represent three key stakeholder domains: governmental policy and enforcement agencies, academic institutions specializing in maritime safety and ocean governance, and frontline user communities with direct operational knowledge. Moreover, the expert panel was deliberately designed to ensure balanced representation across sectors and disciplines, with all participants possessing over ten years of practical or research experience in relevant fields. To further minimize regional bias and enhance the external validity of the study's findings, the selected experts included individuals not only with national-level policy and implementation experience but also with exposure to international safety governance practices, thereby increasing the transferability of consensus outcomes to broader operational contexts, as shown in Table 1.

### Preliminary criteria framework

In this study, preliminary safety management criteria were developed for each of the four dimensions of the SHELL model, drawing upon a comprehensive review of literature pertaining to maritime safety, causal analysis of maritime incidents, and risk factors associated with marine recreational platforms. To enhance the contextual validity and practical relevance of the proposed framework within the Taiwanese coastal recreation environment, in-depth interviews were conducted with 9 experts possessing extensive experience in maritime safety governance, marine leisure platform operations, and ocean

**Table 1. Expert composition table.**

| Expertise Domain | Affiliated | Years of Experience |
|---|---|---|
| Government Agencies | Fleet Branch, CGA, OAC | 18 |
| | Offshore Flotilla 5, Fleet Branch | 22 |
| | Maritime and Port Bureau, MOTC | 11 |
| Academia | Central Police University | 10 |
| | National Taiwan Ocean University | 11 |
| | National Kaohsiung University of Science and Technology | 15 |
| Civil Representatives | Marine Rescue Association | 30 |
| | Recreational Fishing Practitioners | 21 |
| | Powered Platform Users | 15 |

policy. This expert consultation process facilitated the validation of the 4 dimensional structure and the selection of 20 initial criteria tailored to the specific risk landscape of recreational marine activities in Taiwan as shown in Table 2.

The "Software dimension" encompasses the influence of institutional design and procedural frameworks on risk management effectiveness. Specifically, "Comprehensive Management Regulations" examines whether central and local governments have established comprehensive and enforceable regulatory frameworks for marine recreational powered platform management. "Equipment Safety Certification System" assesses equipment compliance with mandatory safety inspection and technical assessment standards. The remaining 3 criteria focus on the institutionalization of safety measures, specifically "Standard Guidelines for Operation ", "Safety Education and Training Courses", and "Regular Inspection Management". Collectively, all 5 criteria within the Software dimension form an essential foundation for operational safety and regulatory management [50–54].The Hardware dimension encompasses the performance and technical conditions of equipment that affect the operational safety of powered platforms. "Platform Functionality and Performance" and "Structural Integrity" address the functionality of mechanical systems and the structural integrity of the platform, both of which are fundamental to navigational safety. "Emergency Rescue Equipment Functionality and Performance" and "Communication and Safety Equipment" focus on the availability and condition of emergency rescue equipment and communication warning systems, which are critical for timely response during accidents. "Regular Inspection and Maintenance" highlights the importance of routine inspection and maintenance in ensuring the continued reliability of all systems. Taken together, these 5 criteria constitute a critical foundation for the assessment and management of safety risks associated with mechanical systems and operational infrastructure [30,50,51,55].

The Environment dimension encompasses key natural and cultural conditions affecting powered platform safety. "Local Maritime Geographic Environment" and "Local Hydrological and Ocean Current Conditions" refer to geographic and hydrological features such as reef distribution, water depth, tides, and currents that influence navigational stability. "Maritime Weather Condition Changes" highlights the importance of responding to sudden weather and sea changes, including

**Table 2. Literature review table.**

| Dimensions | Code | Criteria | Literature Review Source |
|---|---|---|---|
| Software | S1 | Comprehensive Management Regulations | [50–54] |
| | S2 | Equipment Safety Certification System | |
| | S3 | Standard Guidelines for Operation | |
| | S4 | Safety Education and Training Courses | |
| | S5 | Regular Inspection Management | |
| Hardware | H1 | Platform Functionality and Performance | [30,50,51,55] |
| | H2 | Emergency Rescue Equipment Functionality and Performance | |
| | H3 | Communication and Safety Equipment | |
| | H4 | Structural Integrity | |
| | H5 | Regular Inspection and Maintenance | |
| Environmental | E1 | Local Maritime Geographic Environment | [56–59] |
| | E2 | Maritime Weather Condition Changes | |
| | E3 | Local Hydrological and Ocean Current Conditions | |
| | E4 | Marine Traffic Conditions | |
| | E5 | Local Customs and Marine Operation Practices | |
| Liveware | L1 | Emergency Response and Safety Knowledge | [50,51,56,59,60] |
| | L2 | Operational Skills and Proficiency | |
| | L3 | Self-awareness of Physical Condition | |
| | L4 | Marine Safety Awareness | |
| | L5 | Basic Navigation Concepts | |

wind, waves, and visibility. "Marine Traffic Conditions" concerns the density and organization of marine traffic, which may create navigational congestion. "Local Customs and Marine Operation Practices" addresses local customs and traditional marine practices that shape user behavior. Collectively, these 5 criteria reflect the multifaceted environmental risks that must be managed in coastal recreational settings [56–59].

The Liveware dimension encompasses human factors that affect the safe operation of powered platforms. "Emergency Response and Safety Knowledge" and "Marine Safety Awareness" emphasize users' ability to recognize risks, take preventive measures, and respond effectively in emergencies. "Operational Skills and Proficiency " and "Self-awareness of Physical Condition" pertain to operational proficiency and self-assessment of physical and mental readiness to ensure safe handling. "Basic Navigation Concepts" focuses on the understanding and application of basic navigational principles. In combination, these 5 criteria highlight the critical role of individual knowledge, judgment, and skills in ensuring safety within recreational marine environments.

## Results and discussion

This study conducted two rounds of expert questionnaires. The first phase employed a modified Delphi questionnaire, during which the experts' responses were used to calculate the mean value, standard deviation, and coefficient of variation for each criterion. Thereby establishing a representative risk management criteria framework for this study. Based on these results and predefined selection thresholds, the criteria were screened and confirmed for inclusion in the second-phase DEMATEL questionnaire. The final analytical results were through the application of DEMATEL methodology and analysis of the INRM, along with related data analysis and discussion, the correlations among risk criteria were obtained. Comprehensive discussions were conducted across academic, governmental, and civil representative categories, leading to substantive recommendations for future management strategies regarding powered platform risk management.

The research findings indicate that the coefficients of variation (CV) for all criteria factors were less than 0.5, with 95% of criteria factors showing CV below 0.3, as shown in Table 3. This demonstrates that high consensus was achieved in the first round of the MDM investigation during phase one of this study.

This study aimed to identify and prioritize key risk management indicators by conducting expert evaluations of 20 preliminary criteria. Nine domain experts assessed each item in terms of its importance and relevance. As shown in Table 4, the overall mean of the means across all expert responses was 4.04, which exceeds the commonly accepted importance threshold of 4.0 on a five-point Likert-type scale. Based on this result, the study adopted a selection criterion whereby indicators with a mean of means greater than 4.04 and a CV less than or equal to 0.3 were retained as representative factors for subsequent risk management analysis [61,62].

The comprehensive analysis reveals the 5 most critical evaluation criteria factors for powered platform navigation safety risk management, namely: " Emergency Response and Safety Knowledge," "Comprehensive Management Regulations," " Equipment Safety Certification System," "Marine Safety Awareness," and "Local Hydrological and Ocean Current Conditions." Notably, "Comprehensive Management Regulations" and "Equipment Safety Certification System" share identical mean values, jointly ranking second in importance, while "Marine Safety Awareness " and " Local Hydrological and Ocean Current Conditions " also share equal mean values, jointly ranking fourth in importance.

Safety management criteria factors with mean values below 4.0 include 6 factors: "Self-awareness of Physical Condition," "Regular Inspection and Maintenance," "Safety Education and Training Courses," "Regular Inspection Management," "Local Customs and Marine Operation Practices," and "Standard Guidelines for Operation." These factors are considered less significant when evaluating powered platform navigation safety risk management.

Furthermore, criteria factors with mean values equal to 4.0, while still considered very important, include: "Operational Skills and Proficiency," "Platform Functionality and Performance," "Communication and Safety Equipment," and "Local Maritime Geographic Environment." However, as this study aims to conduct analysis using highly representative

**Table 3. MDM investigation statistical analysis.**

| Criteria Code | Mean Value | SD | CV | Ranking |
|---|---|---|---|---|
| S1 | 4.56 | 0.53 | 0.12 | 2 |
| S2 | 4.56 | 0.53 | 0.12 | 2 |
| S3 | 3.33 | 1.00 | 0.3 | 20 |
| S4 | 3.67 | 1.22 | 0.33 | 15 |
| S5 | 3.67 | 1.00 | 0.27 | 15 |
| H1 | 4.00 | 0.71 | 0.18 | 11 |
| H2 | 4.11 | 1.05 | 0.26 | 9 |
| H3 | 4.00 | 1.12 | 0.28 | 11 |
| H4 | 4.11 | 0.60 | 0.15 | 9 |
| H5 | 3.67 | 0.71 | 0.19 | 15 |
| E1 | 4.00 | 0.50 | 0.13 | 11 |
| E2 | 4.22 | 0.44 | 0.10 | 6 |
| E3 | 4.33 | 0.71 | 0.16 | 4 |
| E4 | 4.22 | 0.83 | 0.20 | 6 |
| E5 | 3.56 | 0.53 | 0.15 | 19 |
| L1 | 4.67 | 0.50 | 0.11 | 1 |
| L2 | 4.00 | 0.87 | 0.22 | 11 |
| L3 | 3.67 | 1.00 | 0.27 | 15 |
| L4 | 4.33 | 0.87 | 0.20 | 4 |
| L5 | 4.22 | 0.97 | 0.23 | 6 |
| | **4.04** | | | |

and critically important criteria factors, only those with mean values exceeding 4.04 were selected for the second phase DEMATEL investigation. Fig 1 illustrates the framework of representative criteria factors for powered platform navigation safety risk management in this study.

Through the DEMATEL process, a 10*10 direct relation matrix X is first obtained. Within this matrix, since variables cannot directly compare influence levels with themselves, the diagonal values in matrix X are all 0, as shown in Table 5.

Following the normalization calculations and DEMATEL procedures, the total influence matrix T is obtained, as presented in Table 6. This table demonstrates the interrelationships among the 10 criteria, indicating high levels of correlation and dependency between criteria. This provides decision-makers with a basis for determining management priorities among various criteria to achieve effective safety management outcomes.

By calculating the $d_i$ value, $r_j$ value, centrality $(d_i + r_j)$ value, and net effect degree $(d_i − r_j)$ value for each criteria, we can further elucidate the relative importance among the 10 criteria. According to Table 5, "Emergency Response and Safety Knowledge" exhibits the highest centrality value $(d_i + r_j)$, indicating its substantial core influence on the research subject. "Comprehensive Management Regulations " demonstrates the highest positive net effect degree $(d_i − r_j)$, categorizing it as a cause-type criteria with significant influence over other criteria. Conversely, "Emergency Rescue Equipment Functionality and Performance" shows the lowest net effect degree $(d_i − r_j)$ with a negative value, classifying it as an effect-type criteria that is susceptible to influence from other criteria.

Table 7 presents the centrality and net effect degree values for each criteria factor. These values are plotted on a two-dimensional coordinate system to create a Network Relationship Map, as illustrated in Fig 2. In this figure, the orange vertical line represents the mean centrality value $(d_i + r_j = 12.58)$, while the green horizontal line indicates the mean net effect degree value $(d_i − r_j = 0.00)$. Criteria positioned above the horizontal axis are categorized as cause-type criteria, while those below are classified as effect-type criteria.

**Table 4. Results for the determination of criteria.**

| Criteria Code | Selection Criteria | | Select Results |
|---|---|---|---|
| | Mean Value>4.04 | CV ≤ 0.3 | |
| S1 | 4.56>4.04 | 0.12≦0.3 | Retained |
| S2 | 4.56>4.04 | 0.12≦0.3 | Retained |
| S3 | 3.33<4.04 | 0.3≦0.3 | Removed |
| S4 | 3.67<4.04 | 0.33>0.3 | Removed |
| S5 | 3.67<4.04 | 0.27≦0.3 | Removed |
| H1 | 4.00<4.04 | 0.18≦0.3 | Removed |
| H2 | 4.11>4.04 | 0.26≦0.3 | Retained |
| H3 | 4.00<4.04 | 0.28≦0.3 | Removed |
| H4 | 4.11>4.04 | 0.15≦0.3 | Retained |
| H5 | 3.67<4.04 | 0.19≦0.3 | Removed |
| E1 | 4.00<4.04 | 0.13≦0.3 | Removed |
| E2 | 4.22>4.04 | 0.10≦0.3 | Retained |
| E3 | 4.33>4.04 | 0.16≦0.3 | Retained |
| E4 | 4.22>4.04 | 0.20≦0.3 | Retained |
| E5 | 3.56<4.04 | 0.15≦0.3 | Removed |
| L1 | 4.67>4.04 | 0.11≦0.3 | Retained |
| L2 | 4.00<4.04 | 0.22≦0.3 | Removed |
| L3 | 3.67<4.04 | 0.27≦0.3 | Removed |
| L4 | 4.33>4.04 | 0.20≦0.3 | Retained |
| L5 | 4.22>4.04 | 0.23≦0.3 | Retained |

CV: coefficients of variation.

Based on the Network Relationship Diagram shown in Fig 2, the distribution of key criteria factors for "Powered Platform Navigation Safety Risk Management" can be analyzed as follows:

Top-right quadrant indicates criteria with high centrality and positive influence. "Fundamental navigation concepts" represents the sole criterion positioned within this quadrant, signifying unanimous consensus among all participants regarding its exceptional combination of high influence and high relevance. This unique criterion constitutes the most critical core indicator for the present research domain and warrants classification as the highest priority element. Top-left quadrant represents criteria characterized by low centrality but positive influence. This quadrant includes 3 specific criteria: " Comprehensive Management Regulations," " Equipment Safety Certification System," and " Local Hydrological and Ocean Current Conditions." These criteria are classified as driving needs, indicating that while their influence is relatively low, their high degree of relevance allows them to affect certain other criteria within the framework.

Bottom-left quadrant indicates criteria with low centrality and negative influence. This quadrant comprises 3 specific elements: "Structural Integrity", "Marine Traffic Conditions" and "Emergency Rescue Equipment Functionality and Performance". These criteria demonstrate a high degree of independence and limited relevance to other factors within the system. Given their minimal interaction and negligible impact on other criteria, they are typically addressed through independent or standalone management strategies. Bottom-right quadrant indicates criteria with high centrality and negative influence. This quadrant includes 3 specific criteria: "Marine Safety Awareness", "Emergency Response and Safety Knowledge" and "Maritime Weather Condition Changes". These criteria are considered high-priority for management due to their central systemic role, yet they do not require direct intervention. Instead, their outcomes can be effectively influenced by addressing the interrelated criteria located in the top-right and top-left quadrants. By strengthening or establishing the

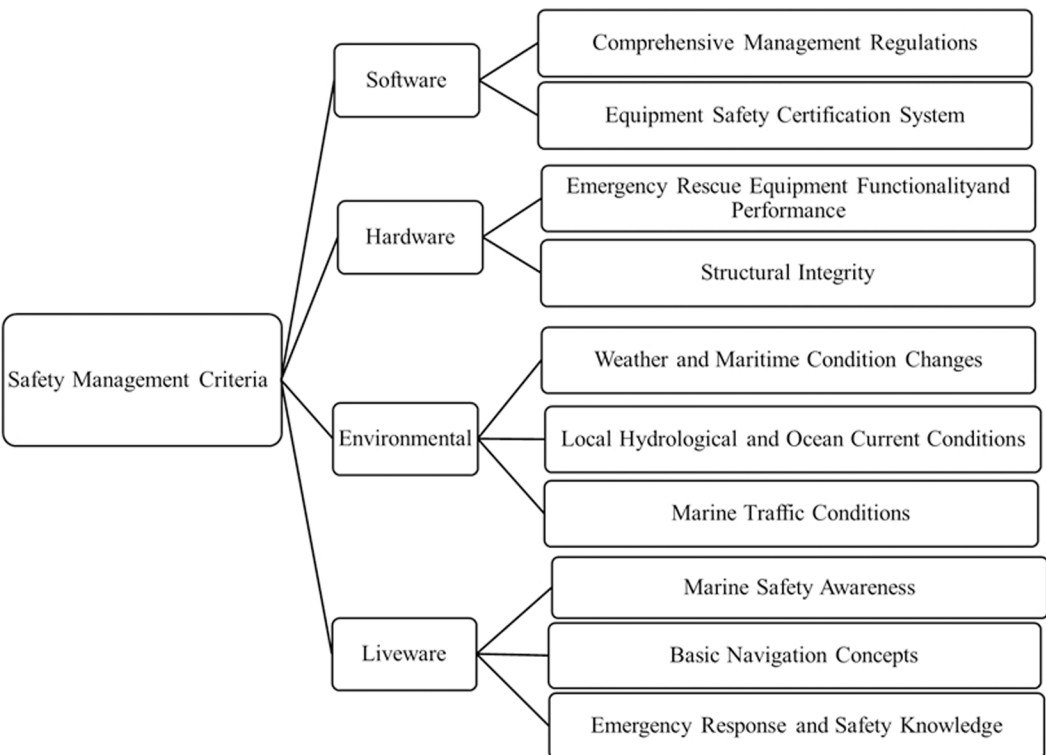

**Fig 1. Safety management criteria framework.**

**Table 5. Direct relation matrix X.**

| X | S1 | S2 | H2 | H4 | E2 | E3 | E4 | L1 | L4 | L5 | Total |
|---|----|----|----|----|----|----|----|----|----|----|-------|
| S1 | 0.00 | 3.11 | 2.78 | 3.11 | 1.44 | 1.67 | 3.00 | 2.22 | 2.33 | 2.67 | 22.33 |
| S2 | 2.33 | 0.00 | 2.89 | 3.11 | 2.33 | 1.67 | 2.33 | 2.44 | 2.11 | 1.67 | 20.89 |
| H2 | 1.67 | 2.44 | 0.00 | 1.33 | 1.56 | 1.67 | 1.56 | 3.22 | 2.00 | 1.44 | 16.89 |
| H4 | 1.67 | 2.78 | 1.67 | 0.00 | 2.22 | 1.33 | 1.78 | 1.78 | 1.33 | 1.56 | 16.11 |
| E2 | 1.00 | 2.00 | 2.44 | 2.33 | 0.00 | 3.67 | 2.44 | 2.56 | 2.56 | 2.33 | 21.33 |
| E3 | 1.67 | 1.67 | 2.11 | 1.56 | 3.33 | 0.00 | 2.67 | 2.44 | 2.56 | 2.56 | 20.56 |
| E4 | 2.00 | 1.78 | 1.56 | 1.11 | 2.56 | 2.00 | 0.00 | 2.11 | 2.44 | 3.11 | 18.67 |
| L1 | 2.00 | 2.11 | 3.00 | 1.78 | 2.67 | 2.44 | 2.11 | 0.00 | 3.67 | 2.89 | 22.67 |
| L4 | 2.33 | 1.44 | 1.67 | 1.56 | 2.89 | 2.44 | 2.56 | 3.67 | 0.00 | 3.11 | 21.67 |
| L5 | 2.11 | 1.78 | 2.56 | 1.89 | 2.67 | 2.44 | 2.67 | 3.22 | 3.11 | 0.00 | 22.44 |

four key criteria within those quadrants, a cascading effect can be achieved, indirectly improving the performance of the bottom-right quadrant criteria.

To conduct an in-depth analysis of the inter-relationships among criteria factors, this study established a significance threshold value of 0.63, representing the mean value of criteria factors. The purpose of setting this threshold value is to eliminate criteria with less significant correlations and facilitate further interpretation of the relationships and mutual influences among criteria factors.

**Table 6. Total-influence matrix T.**

| T | S1 | S2 | H2 | H4 | E2 | E3 | E4 | L1 | L4 | L5 | di |
|---|-----|-----|-----|-----|-----|-----|-----|-----|-----|-----|-----|
| S1 | 0.50 | 0.67 | 0.70 | 0.63 | 0.69 | 0.63 | 0.72 | 0.77 | 0.73 | 0.72 | 6.76 |
| S2 | 0.55 | 0.52 | 0.67 | 0.60 | 0.68 | 0.60 | 0.66 | 0.73 | 0.69 | 0.65 | 6.34 |
| H2 | 0.46 | 0.53 | 0.47 | 0.46 | 0.56 | 0.52 | 0.54 | 0.66 | 0.59 | 0.55 | 5.33 |
| H4 | 0.43 | 0.52 | 0.51 | 0.39 | 0.56 | 0.48 | 0.52 | 0.58 | 0.54 | 0.52 | 5.05 |
| E2 | 0.52 | 0.61 | 0.67 | 0.58 | 0.61 | 0.69 | 0.68 | 0.76 | 0.72 | 0.69 | 6.53 |
| E3 | 0.53 | 0.58 | 0.65 | 0.54 | 0.72 | 0.54 | 0.68 | 0.74 | 0.71 | 0.69 | 6.38 |
| E4 | 0.51 | 0.55 | 0.58 | 0.49 | 0.65 | 0.58 | 0.53 | 0.68 | 0.66 | 0.66 | 5.89 |
| L1 | 0.59 | 0.64 | 0.73 | 0.60 | 0.75 | 0.68 | 0.71 | 0.71 | 0.80 | 0.75 | 6.96 |
| L4 | 0.58 | 0.60 | 0.67 | 0.57 | 0.74 | 0.67 | 0.71 | 0.82 | 0.65 | 0.74 | 6.75 |
| L5 | 0.59 | 0.63 | 0.71 | 0.60 | 0.75 | 0.68 | 0.72 | 0.82 | 0.78 | 0.64 | 6.90 |
| rj | 5.25 | 5.84 | 6.36 | 5.46 | 6.70 | 6.08 | 6.47 | 7.26 | 6.86 | 6.61 | |

**Table 7. The values of the centrality ($d_i+r_j$) and net effect ($d_i-r_j$).**

| Criteria Code | $d_i$ | $r_j$ | $d_i + r_j$ | $d_i - r_j$ |
|---|-----|-----|-----|-----|
| S1 | 6.76 | 5.25 | 12.01 | 1.50 |
| S2 | 6.34 | 5.84 | 12.18 | 0.51 |
| H2 | 5.33 | 6.36 | 11.70 | −1.03 |
| H4 | 5.05 | 5.46 | 10.50 | −0.41 |
| E2 | 6.53 | 6.70 | 13.23 | −0.17 |
| E3 | 6.38 | 6.08 | 12.46 | 0.30 |
| E4 | 5.89 | 6.47 | 12.37 | −0.58 |
| L1 | 6.96 | 7.26 | 14.22 | −0.31 |
| L4 | 6.75 | 6.86 | 13.61 | −0.11 |
| L5 | 6.90 | 6.61 | 13.51 | 0.29 |
| | | | 12.58 | 0 |

In Fig 3, arrows indicate the directional influence between criteria factors. Solid double-headed arrows represent bidirectional influences between criteria factors, while dashed single-headed arrows indicate unidirectional influence from one criteria factor to another. Among the key criteria factors for "Powered Platform Navigation Safety Risk Management," four factors belong to the cause group that directly influence other criteria factors: "Comprehensive Management Regulations," "Equipment Safety Certification System," "Basic Navigation Concepts," and "Local Hydrological and Ocean Current Conditions."

Regarding "Comprehensive Management Regulations," this criteria factor influences seven other criteria: "Emergency Response and Safety Knowledge," "Marine Safety Awareness," "Basic Navigation Concepts," " Emergency Rescue Equipment Functionality and Performance," "Equipment Safety Certification System," "Maritime Weather Condition Changes," and " Marine Traffic Conditions." This demonstrates that establishing comprehensive management regulations enables effective management of these seven criteria. Thus, the primary step in powered platform risk management is establishing appropriate regulations for the platform itself and its various water activities, including comprehensive central and local government management frameworks and safety certification standards [63–65]. Establishing comprehensive integrated management regulations and safety certification standards by central and local governments enables undifferentiated regulatory management, providing clear guidelines for public compliance. Additionally, it strengthens regulatory enforcement and increases the costs of non-compliance, creating a safer and more reliable marine recreational environment for users.

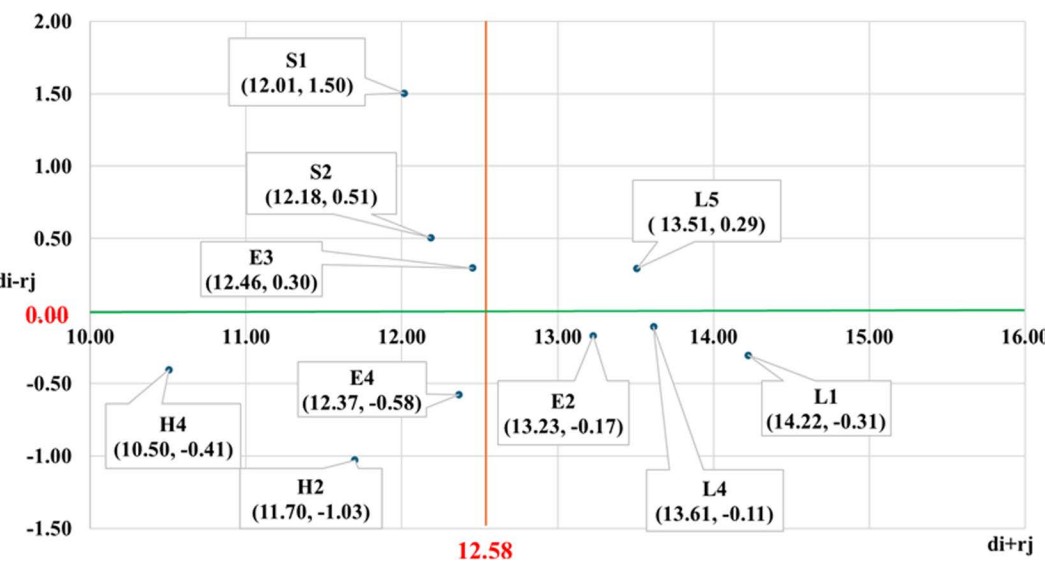

**Fig 2. Influence network relation map.**

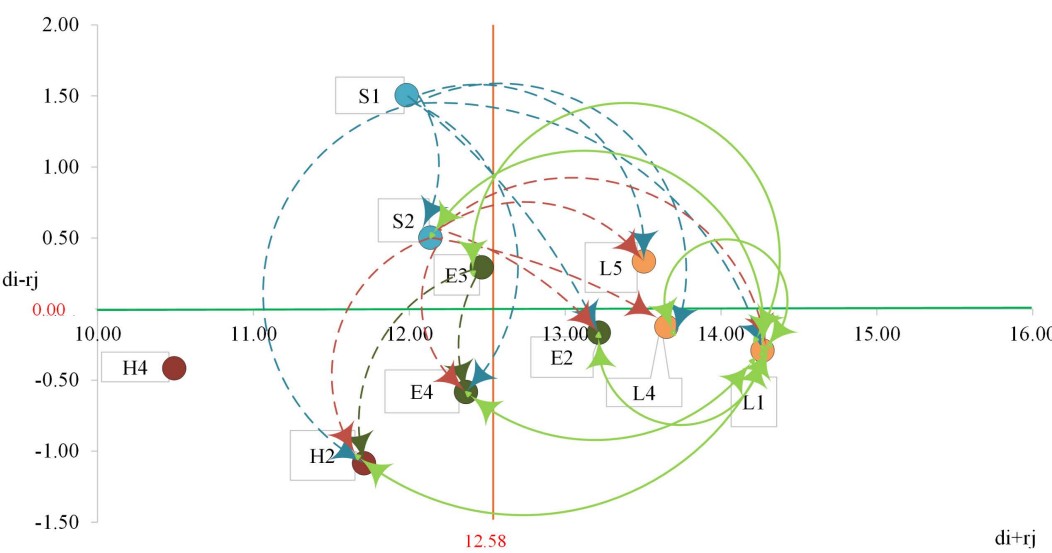

**Fig 3. Criteria relationship map.**

For "Equipment Safety Certification System," this criteria factor influences 6 criteria: "Emergency Response and Safety Knowledge," "Marine Safety Awareness," "Basic Navigation Concepts," " Emergency Rescue Equipment Functionality and Performance," " Maritime Weather Condition Changes," and " Marine Traffic Conditions." This implies that establishing product safety certification procedures, including product testing and durability safety standard assessments [66–68], can enhance management of these criteria factors. The establishment of product safety certification procedures, including product testing and durability safety standard assessment, can comprehensively enhance the management standards of maritime recreational motorized platforms. In the long term, a well-developed equipment safety certification system ensures that marine recreational powered platforms circulating in the market meet minimum safety requirements,

improving equipment reliability and service life, reducing incidents caused by equipment failure, and providing users with safer maritime recreational tools.

Concerning "Basic Navigation Concepts," this criteria factor influences 6 criteria: "Emergency Response and Safety Knowledge," "Marine Safety Awareness," " Emergency Rescue Equipment Functionality and Performance," " Maritime Weather Condition Changes," "Local Hydrological and Ocean Current Conditions," and " Marine Traffic Conditions." Establishing fundamental navigation knowledge for users, including basic navigation rules, collision avoidance rules, and navigation mark identification [69–70]. The establishment of fundamental navigational knowledge among users is crucial for enhancing the long-term safety of marine recreational powered platforms. This foundational navigational knowledge not only assists users in preventing accidents but also enables them to make appropriate judgments during hazardous situations, thereby reducing the risk of casualties. It significantly elevates users' risk awareness and safety consciousness regarding safety standards, comprehensively strengthening navigational safety and ensuring the sustainable development of maritime recreational activities.

Regarding "Local Hydrological and Ocean Current Conditions," this criteria factor influences six criteria: "Emergency Response and Safety Knowledge," "Marine Safety Awareness," "Basic Navigation Concepts," "Emergency Rescue Equipment Functionality and Performance," "Maritime Weather Condition Changes," and " Marine Traffic Conditions." When users are adequately informed about hydrological factors affecting navigation, such as tides, ocean currents, and water temperature [71,72], this knowledge enhances navigation safety management across these six criteria. Additionally, through marine spatial planning [73,74], recreational use spaces can be regulated to ensure maritime safety. For instance, the delineation of exclusive marine recreational zones that are spatially segregated from fishing grounds and commercial navigation routes, combined with the implementation of rescue monitoring systems and emergency response infrastructure within these designated areas, can significantly mitigate the risk of inter-user conflicts and maritime accidents. Such spatial planning not only enhances the personal safety of recreational users but also contributes to the sustainable management of marine resources and facilitates the multifunctional and equitable use of ocean space.

Drawing upon the findings of this study, recent incidents provide compelling evidence of the practical implications of structural standards and operator competence in enhancing navigational safety. For instance, in January 2024, four anglers operated 2 powered platforms for offshore fishing. One platform lost propulsion after taking on water under rough sea conditions and was subsequently capsized when waves pushed it against a breakwater. This incident suggests that, had the platform's structural integrity been assured through certified quality standards and the operators possessed fundamental maritime safety knowledge, the risk of capsizing could have been significantly mitigated.

A similar incident occurred in April 2024, involving two anglers who were thrown overboard after their powered platform capsized while at sea. This case underscores the potential value of real-time coastal radar surveillance and enhanced platform seaworthiness. Improved structural resilience and timely monitoring capabilities could have strengthened the platform's ability to withstand adverse sea conditions and ensured a more effective emergency response.

These cases collectively reinforce the need for a dual emphasis on platform design certification and safety-oriented user education as foundational elements of a robust recreational marine safety management framework.

In examining the framework for managing the navigational safety of marine recreational powered platforms, this study identifies several potential challenges. These include imbalanced resource allocation, inconsistent technical standards, limited monitoring capacity, and difficulties in user education.

• The unequal distribution of safety management resources between central and local governments may lead to inconsistencies in safety standards and administrative enforcement across different regions. Inter-agency coordination further poses a significant challenge, potentially reducing the overall efficiency of policy implementation.

• Given the diverse types and configurations of marine recreational powered platforms, establishing a unified and applicable safety certification standard presents substantial technical difficulties. Additionally, industry stakeholders may resist such regulatory measures due to cost concerns, further complicating enforcement.

- When these platforms operate in more remote offshore areas, their small size and distant location may hinder real-time tracking by coastal radar systems, resulting in delayed emergency response and increased rescue difficulty.

- Participants in recreational marine activities often lack safety awareness or show resistance toward mandatory safety requirements, thereby posing obstacles to the cultivation of a safety-oriented culture in the marine leisure sector.

This study contributes to the advancement of existing scholarship by applying the SHELL model to a previously underexplored context, specifically the safety management of marine recreational powered platforms. While earlier research has primarily focused on the application of the SHELL framework within aviation and structured maritime operations, its use in decentralized environments that involve individual users and informal practices remains limited. This is particularly true when the model is combined with expert judgment approaches for evaluating complex criteria.

By adjusting the SHELL model to reflect the realities of informal operator behavior, loosely defined regulatory structures, and highly variable environmental conditions, this research offers new theoretical perspectives on how individuals, procedures, physical systems, and natural surroundings interact within hazardous recreational settings. The empirical results underscore the essential role of the relationship between personnel and procedural components, especially in areas such as insufficient safety training, weak enforcement mechanisms, and inconsistent levels of user knowledge. These issues are often overlooked in studies focused on formal commercial maritime systems.

Through a critical evaluation of its academic contribution, this study demonstrates that combining the SHELL framework with expert-based evaluation methods, including structured group consultation and relationship modeling, improves both the identification of influential risk factors and the ability to prioritize them under conditions where resources are limited. This methodological approach transforms the SHELL model from a descriptive framework into a practical system that supports decision making and guides the development of adaptive marine safety policies. In doing so, the research enriches theoretical understanding and provides applicable insights for other maritime sectors that lack formal regulation.

This study conducts a comparative analysis of the proposed SHELL-based framework alongside other established safety models and regional regulatory systems. From a theoretical perspective, the Human Factors Analysis and Classification System (HFACS) has been widely applied in postincident investigations across structured domains such as aviation and military operations. However, its foundational assumptions, including the presence of formal supervisory mechanisms, hierarchical decision-making processes, and standardized operational procedures, are typically not present in general maritime recreational environments. In such settings, users of marine recreational powered platforms often operate independently and without formal regulatory oversight. In contrast, the adapted SHELL framework developed in this study emphasizes dynamic analysis of operational interfaces, enabling forward-looking risk evaluation under conditions characterized by procedural flexibility and uncertain governance.

At the level of regulatory practice, Japan's framework under the Small Vessel Registration Ordinance classifies platforms shorter than three meters in length and with engine power below 1.5 kilowatts as mini-vessels. Although these platforms are exempt from registration and formal safety inspections, Japan promotes basic operator knowledge through the distribution of a "Mini-Vessel Safety Manual," which encourages familiarity with navigation rules and essential engine operation. In South Korea, the Water Leisure Safety Act provides a legislative basis for maintaining the safety and order of water-based recreational activities. These systems depend on strong institutional capacity and widespread public adherence to regulations, conditions that are often not yet established in jurisdictions where safety governance remains in an early stage of development.

In contrast, Taiwan's current approach to maritime recreational safety is characterized by fragmented oversight, particularly in areas such as license administration, enforcement coordination, and user education. To respond to these challenges, the SHELL-based framework proposed in this study offers a flexible and expandable tool for identifying critical safety risks, allocating limited regulatory resources, and supporting the gradual evolution of policy. Its modular structure allows for phased implementation in settings where institutional capacity is still being developed. Moreover, the analytical

model can be extended to other contexts where individual autonomy and limited oversight coincide, such as personal watercraft use and adventure tourism.

This comparative assessment highlights the distinct value of the present study by demonstrating that a conceptually grounded and practically adaptable framework can contribute meaningfully to safety governance in environments that do not align with traditional safety model assumptions. In doing so, the study enhances maritime safety theory and provides an actionable tool for policy development in emerging and insufficiently regulated operational domains.

The analytical framework developed in this study integrates the SHELL model with the Delphi method and DEMATEL analysis. It offers considerable flexibility in addressing challenges associated with emerging maritime equipment and non-traditional vessels that currently fall outside comprehensive regulatory oversight. This approach can be extended to Pacific Island nations such as Fiji and Palau, where marine tourism is highly active yet regulatory systems remain under-developed, reflecting conditions similar to those in Taiwan. Implementing preventive measures at an early stage is critical to mitigating the escalation of systemic risks [75–77]. Notably, Palau has effectively achieved its marine management and conservation objectives through a high level of local community and stakeholder engagement [78]. In addition, the pro-posed framework is relevant to other island countries in Asia, such as the Philippines. Similar to Taiwan, the Philippines is often affected by typhoons and strong ocean currents like the Kuroshio Current. These environmental challenges highlight the urgent need to strengthen maritime safety systems and improve the regulation of motorized recreational vessels, in order to reduce risks and better protect public safety and property [78]. Moreover, both Taiwan and the Philippines face significant challenges in integrated management and are subject to high levels of risk pressure associated with marine recreational activities [79,80]. The methodology supports the formulation of structured safety certification systems and coordination mechanisms among stakeholders. By adjusting key contextual variables such as stakeholder roles, environ-mental conditions, and institutional frameworks, this framework contributes to strategic safety management and policy development in comparable settings. Such applications also allow for the validation of its generalizability and robustness across diverse maritime governance environments.

## Conclusions

In recent years, the rapid expansion of marine tourism has resulted in a marked increase in the use of powered recre-ational platforms by tourists, which has, in turn, underscored the pressing need for effective maritime safety governance. To address this emerging issue, this study introduces an innovative analytical framework that combines the SHELL model with the Delphi method and DEMATEL analysis. This framework provides both a structured and systematic foundation and the adaptability required to meet the demands of diverse administrative environments. It facilitates the formulation of a comprehensive set of safety management criteria tailored to the characteristics of powered recreational platforms. For governmental agencies and maritime regulators, the outcomes of this study serve as a reference for prioritizing key criteria and interpreting their interrelationships. These insights support the design of focused policy interventions, including the formulation of unified regulatory measures, the implementation of foundational navigation training programs, and the allocation of specific maritime areas to strengthen oversight responsibilities.

Viewed holistically, the proposed methodology and recommendations demonstrate a coherent integration of theoreti-cal precision and practical utility. Although developed within the context of Taiwan's maritime regulatory environment, the framework possesses considerable potential for application in other coastal and island nations that encounter comparable institutional constraints. Moreover, the findings expose systemic gaps between policy formulation and practical execution, as evidenced through real incidents. As illustrated in the 2024 case, measures such as promoting public understanding of basic navigational knowledge, restricting leisure activities to designated maritime areas, and introducing safety surveil-lance systems for recreational users may collectively enhance maritime safety. These efforts contribute to the timely dis-semination of warning information and support the operational management of coastal activities and emergency response personnel.

From a comprehensive perspective, this study offers a representative and integrative methodological approach with high relevance to both academic inquiry and policy implementation. It provides a valuable reference for improving the governance of recreational maritime vessels, particularly in jurisdictions where regulatory structures are incomplete or remain in a fragmented state.

## Supporting information

**S1 File. Survey data and result analysis for each criterion from 9 experts.**
(DOCX)

**S2 File. DEMATEL questionnaire.**
(DOCX)

## Author contributions

**Conceptualization:** Shao-Hua Hsu, Yo-Kang Yang, Meng-Tsung Lee, Jao-Chuan Lin.

**Data curation:** Shao-Hua Hsu, Yo-Kang Yang, Ya-Fan Ho.

**Formal analysis:** Ya-Fan Ho, Meng-Tsung Lee, Jao-Chuan Lin.

**Investigation:** Shao-Hua Hsu, Yo-Kang Yang.

**Methodology:** Shao-Hua Hsu, Ya-Fan Ho, Meng-Tsung Lee, Jao-Chuan Lin.

**Resources:** Shao-Hua Hsu.

**Validation:** Shao-Hua Hsu, Meng-Tsung Lee.

**Visualization:** Shao-Hua Hsu, Jao-Chuan Lin.

**Writing – original draft:** Shao-Hua Hsu, Yo-Kang Yang, Ya-Fan Ho, Jao-Chuan Lin.

**Writing – review & editing:** Shao-Hua Hsu, Jao-Chuan Lin.

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
