## [Decision Letter · Decision Letter 0]

27 May 2025

PONE-D-25-20340A Multi-Criteria Decision-Making Framework for Managing the Safety of Marine Recreational Powered Platforms: Integration with the SHELL ModelPLOS ONE

Dear Dr. Hsu,

Thank you for submitting your manuscript to PLOS ONE. After careful consideration, we feel that it has merit but does not fully meet PLOS ONE’s publication criteria as it currently stands. Therefore, we invite you to submit a revised version of the manuscript that addresses the points raised during the review process.

We look forward to receiving your revised manuscript.

Kind regards,

Yi-Che Shih, Ph.D.

Academic Editor

PLOS ONE

Journal Requirements:

3. We note you have included a table to which you do not refer in the text of your manuscript. Please ensure that you refer to Table 3 in your text; if accepted, production will need this reference to link the reader to the Table.

Additional Editor Comments :

We have now completed the reviewing process of your article PONE-D-25-20340 entitled ""A Multi-Criteria Decision-Making Framework for Managing the Safety of Marine Recreational Powered Platforms: Integration with the SHELL Model"

submitted to the PLOS One.

According to the reviewers' comments, this manuscript needs major revision before consideration for acceptance.

Please read the reviewers' recommendations listed below and revise your article in light of their comments.

We look forward to receiving your resubmission soon.

Reviewer #1

This study focuses on marine recreational activities, addressing a critical gap in navigational safety management. The integration of the SHELL MODEL’s four-dimensional framework ("Software," "Hardware," "Environment," and "Liveware") demonstrates interdisciplinary rigor. However, several methodological and theoretical refinements are recommended to strengthen the manuscript’s validity and impact.

1. While the SHELL MODEL is well-established in safety science, the manuscript must explicitly justify the selection of specific indicators under each dimension. For instance, how were the 20 initial criteria derived, and why were these 10 prioritized? A detailed explanation of the framework’s completeness and credibility is essential.

2. The Modified Delphi and DEMATEL processes lack clarity in key methodological details. For example: How were the 10 experts selected? What theoretical or empirical bases informed the consensus thresholds for criterion retention? How were dissenting opinions resolved?

3. The reliance on traditional multi-criteria decision analysis (MCDA) methodslimits the study’s novelty. To advance the field, the authors should: Critically assess how their approach extends existing literature. Discuss theoretical implications (e.g., how the findings reshape SHELL MODEL applications in maritime safety).

4. The manuscript needs to conduct comparative discussions with alternative safety frameworks or regional regulatory systems. Including such analyses would highlight the study’s unique value proposition and contextualize its recommendations within broader safety management paradigms.

Reviewer #2

• The study applies an innovative integration of the SHELL model and DEMATEL within a marine safety research field. However, there are still some areas that need improvement before acceptance.

1. The abstract part is repetitive phrasing, especially with "powered platform" and overly dense technical jargon. It is recommended that the author simplify and condense the abstract for accessibility.

2. Although this article provides a good overview of SHELL and maritime risk, the exact research gap (i.e., why SHELL+MCDM is more effective for powered platforms than HFACS or existing frameworks) needs more precise articulation. It is suggested that the author could explicitly contrast the limitations of prior models like HFACS and state why SHELL is better suited for recreational risk environments.

3. The manuscript cites a significant number of references, but these are not sufficiently integrated to support the research logic and discussions. It is recommended that the authors integrate Table 2 more effectively into the main text. For example, clarify how each dimension (software, hardware, etc.) is justified based on the cited references. Use subheadings or summary paragraphs to guide readers in understanding the standard-setting process.

4. On page 11, the authors mentioned total of 9 participants were selected from different fields. For a highly heterogeneous topic, this number is relatively small. It is recommend that the authors explain why 9 participants are sufficient to reach a consensus and address potential biases (such as local vs. international applicability).

5. Pages 19, 20. The author shows that Figures 2 and 3 (INRM & Criteria Relationship Map) are important but not clearly explained. It is suggested that the author could add brief figure captions explaining the quadrant's meaning and how practitioners can interpret them. For example: "Top-right quadrant indicates criteria with high centrality and positive influence…"

6. For the journal market and its international readership, discussions and case studies of this manuscript are limited to Taiwan and may restrict broader impact. Authors are encouraged to briefly discuss how this framework could be adapted to other countries facing similar regulatory gaps, such as those in Southeast Asia or the Pacific Islands.

7. On pages 25-26 the conclusion part. The conclusion repeats the research findings but fails to highlight new content and its significant contributions. It is recommended that the authors add a summary of the findings, such as the innovative integration of SHELL and DEMATEL, a practical tool for policymakers in leisure marine safety, validation of recommendations using empirical events, etc.

8. Before considering accepting the article, it is recommended that some revisions be made.

Reviewers' comments:

Reviewer's Responses to Questions

**Comments to the Author**

1. Is the manuscript technically sound, and do the data support the conclusions?

Reviewer #1: Yes

Reviewer #2: Partly

2. Has the statistical analysis been performed appropriately and rigorously? 

Reviewer #1: Yes

Reviewer #2: Yes

3. Have the authors made all data underlying the findings in their manuscript fully available?

Reviewer #1: Yes

Reviewer #2: Yes

4. Is the manuscript presented in an intelligible fashion and written in standard English?

Reviewer #1: Yes

Reviewer #2: Yes

5. Review Comments to the Author

Reviewer #1: This study focuses on marine recreational activities, addressing a critical gap in navigational safety management. The integration of the SHELL MODEL’s four-dimensional framework ("Software," "Hardware," "Environment," and "Liveware") demonstrates interdisciplinary rigor. However, several methodological and theoretical refinements are recommended to strengthen the manuscript’s validity and impact.

1. While the SHELL MODEL is well-established in safety science, the manuscript must explicitly justify the selection of specific indicators under each dimension. For instance, how were the 20 initial criteria derived, and why were these 10 prioritized? A detailed explanation of the framework’s completeness and credibility is essential.

2. The Modified Delphi and DEMATEL processes lack clarity in key methodological details. For example: How were the 10 experts selected? What theoretical or empirical bases informed the consensus thresholds for criterion retention? How were dissenting opinions resolved?

3. The reliance on traditional multi-criteria decision analysis (MCDA) methodslimits the study’s novelty. To advance the field, the authors should: Critically assess how their approach extends existing literature. Discuss theoretical implications (e.g., how the findings reshape SHELL MODEL applications in maritime safety).

4. The manuscript needs to conduct comparative discussions with alternative safety frameworks or regional regulatory systems. Including such analyses would highlight the study’s unique value proposition and contextualize its recommendations within broader safety management paradigms.

Reviewer #2: • The study applies an innovative integration of the SHELL model and DEMATEL within a marine safety research field. However, there are still some areas that need improvement before acceptance.

1. The abstract part is repetitive phrasing, especially with "powered platform" and overly dense technical jargon. It is recommended that the author simplify and condense the abstract for accessibility.

2. Although this article provides a good overview of SHELL and maritime risk, the exact research gap (i.e., why SHELL+MCDM is more effective for powered platforms than HFACS or existing frameworks) needs more precise articulation. It is suggested that the author could explicitly contrast the limitations of prior models like HFACS and state why SHELL is better suited for recreational risk environments.

3. The manuscript cites a significant number of references, but these are not sufficiently integrated to support the research logic and discussions. It is recommended that the authors integrate Table 2 more effectively into the main text. For example, clarify how each dimension (software, hardware, etc.) is justified based on the cited references. Use subheadings or summary paragraphs to guide readers in understanding the standard-setting process.

4. On page 11, the authors mentioned total of 9 participants were selected from different fields. For a highly heterogeneous topic, this number is relatively small. It is recommend that the authors explain why 9 participants are sufficient to reach a consensus and address potential biases (such as local vs. international applicability).

5. Pages 19, 20. The author shows that Figures 2 and 3 (INRM & Criteria Relationship Map) are important but not clearly explained. It is suggested that the author could add brief figure captions explaining the quadrant's meaning and how practitioners can interpret them. For example: "Top-right quadrant indicates criteria with high centrality and positive influence…"

6. For the journal market and its international readership, discussions and case studies of this manuscript are limited to Taiwan and may restrict broader impact. Authors are encouraged to briefly discuss how this framework could be adapted to other countries facing similar regulatory gaps, such as those in Southeast Asia or the Pacific Islands.

7. On pages 25-26 the conclusion part. The conclusion repeats the research findings but fails to highlight new content and its significant contributions. It is recommended that the authors add a summary of the findings, such as the innovative integration of SHELL and DEMATEL, a practical tool for policymakers in leisure marine safety, validation of recommendations using empirical events, etc.

8. Before considering accepting the article, it is recommended that some revisions be made.

6. PLOS authors have the option to publish the peer review history of their article (what does this mean? ). If published, this will include your full peer review and any attached files.

**Do you want your identity to be public for this peer review?** For information about this choice, including consent withdrawal, please see our Privacy Policy .

Reviewer #1: No

Reviewer #2: No

---

## [Author Response · Author response to Decision Letter 1]

23 Jun 2025

Response to Reviewers

Reviewer #1

This study focuses on marine recreational activities, addressing a critical gap in navigational safety management. The integration of the SHELL MODEL’s four-dimensional framework ("Software," "Hardware," "Environment," and "Liveware") demonstrates interdisciplinary rigor. However, several methodological and theoretical refinements are recommended to strengthen the manuscript’s validity and impact.

Authors’ Reply: We sincerely appreciate the reviewer’s positive recognition of the study’s thematic focus and the interdisciplinary integration of its methodological framework. We will revise the manuscript accordingly in response to each point, with the aim of further enhancing the scholarly significance and practical relevance of the research findings. Detailed responses to the specific suggestions are provided below.

1. While the SHELL MODEL is well-established in safety science, the manuscript must explicitly justify the selection of specific indicators under each dimension. For instance, how were the 20 initial criteria derived, and why were these 10 prioritized? A detailed explanation of the framework’s completeness and credibility is essential.

Authors’ Reply: Thanks for the valuable comment. In response, we have revised and expanded the section concerning the initial indicator framework (please refer to the revised manuscript, pages 20 to 23, 26). The detailed explanation is as follows

(1) Derivation of the 20 Initial Indicators: (please refer to the revised manuscript, page 20)

This study conducted a comprehensive literature review focusing on maritime safety, causal analysis of maritime accidents, and risk factors associated with recreational maritime platforms. Based on the attributes of each SHELL dimension, 20 preliminary criteria were systematically compiled to reflect the relevant risk domains.

(2) Selection Basis for the 10 Core Indicators: (please refer to the revised manuscript, page 26)

A Delphi survey was conducted with nine experts, each possessing over 10 years of experience in maritime safety, marine law enforcement, and ocean governance practices. Indicators with a mean score greater than 4.05 and a coefficient of variation less than 0.3 were selected as the most representative 10 indicators for subsequent DEMATEL analysis.

(3) Explanation of Framework Completeness and Credibility: (please refer to the revised manuscript, pages 21 to 23)

We have added a detailed explanation of the core criteria selected under each of the four SHELL dimensions. In addition, expert feedback confirmed the rationality, criticality, completeness, and credibility of the selected indicators, thereby reinforcing the validity of the overall analytical framework.

2. The Modified Delphi and DEMATEL processes lack clarity in key methodological details. For example: How were the 10 experts selected? What theoretical or empirical bases informed the consensus thresholds for criterion retention? How were dissenting opinions resolved?

Authors’ Reply: Thanks for the valuable comment. Our detailed responses are as follows: (please refer to the revised manuscript, page 18, 25, 26)

(1) Expert Selection Process: (please refer to the revised manuscript, page 18)

This study employed purposive sampling to recruit a representative panel of experts across key domains, including maritime safety enforcement, maritime governance, search and rescue operations, and user engagement. All experts possessed over 10 years of practical or research experience in their respective fields, ensuring both disciplinary depth and cross-sectoral breadth. To ensure balanced input, three experts were selected from each of the following sectors: government agencies, academia, and civil society, resulting in a total of nine panelists.

(2) Consensus Threshold Criteria: (please refer to the revised manuscript, page 26)

To ensure that the retained criteria reflected both relevance and consensus, indicators with a mean score below 4.05 and a coefficient of variation (CV) greater than 0.3 were excluded. These thresholds were adopted to capture both the central tendency and the degree of dispersion in expert evaluations.

(3) Handling of Divergent Opinions: (please refer to the revised manuscript, page 25)

After the first round of the Delphi process, results indicated high consistency across responses: all criteria had a CV lower than 0.5, and 95% had a CV below 0.3, suggesting a strong level of agreement. Nevertheless, in the event of significant divergence among expert opinions, a second Delphi round would have been conducted to refine judgments and reach consensus, in alignment with the iterative feedback mechanism of the Delphi method.

3. The reliance on traditional multi-criteria decision analysis (MCDA) methodslimits the study’s novelty. To advance the field, the authors should: Critically assess how their approach extends existing literature. Discuss theoretical implications (e.g., how the findings reshape SHELL MODEL applications in maritime safety).

Authors’ Reply: Thanks for the valuable comment. In response to this suggestion, we have revised and expanded the discussion and conclusion sections of the manuscript (please refer to the revised manuscript, pages 41-42). These revisions include an extended review of relevant literature and a critical assessment of how the integration of the SHELL model with the Delphi and DEMATEL methods contributes to the development of an innovative framework for maritime safety management.

4. The manuscript needs to conduct comparative discussions with alternative safety frameworks or regional regulatory systems. Including such analyses would highlight the study’s unique value proposition and contextualize its recommendations within broader safety management paradigms.

Authors’ Reply: Thanks for the valuable comment. In response to this comment, we have revised the discussion section to include additional content (please refer to the revised manuscript, pages 43 to 45). Specifically, we added a comparative analysis between the management framework proposed in this study, which integrates the SHELL model, the MDM method, and the DEMATEL technique, and the Human Factors Analysis and Classification System (HFACS). Furthermore, we have highlighted the practical applicability and potential of this framework in regions with insufficient regulatory mechanisms or fragmented maritime safety governance.

Reviewer #2

• The study applies an innovative integration of the SHELL model and DEMATEL within a marine safety research field. However, there are still some areas that need improvement before acceptance.

Authors’ Reply: Thanks for the valuable comment. We sincerely appreciate the reviewer’s positive recognition of the integration of the SHELL model with the MDM and DEMATEL methodologies. In response, we have carefully addressed each comment with corresponding revisions aimed at enhancing the constructiveness and practical relevance of the study. Detailed responses to the specific suggestions are provided below.

1. The abstract part is repetitive phrasing, especially with "powered platform" and overly dense technical jargon. It is recommended that the author simplify and condense the abstract for accessibility.

Authors’ Reply: Thanks for the valuable comment. We have thoroughly revised the abstract to reduce redundancy and improve wording. In addition, the content has been reorganized to enhance conciseness and clarity, while ensuring that the core contributions of the study are accurately preserved.

2. Although this article provides a good overview of SHELL and maritime risk, the exact research gap (i.e., why SHELL+MCDM is more effective for powered platforms than HFACS or existing frameworks) needs more precise articulation. It is suggested that the author could explicitly contrast the limitations of prior models like HFACS and state why SHELL is better suited for recreational risk environments.

Authors’ Reply: Thanks for the valuable comment. In response to this comment, we have revised the introduction and discussion sections (please refer to the revised manuscript, pages 43 to 45) to elaborate on the limitations of alternative frameworks such as HFACS, and to further clarify the rationale for adopting the integrated approach of the SHELL model with MDM and MCDM methods, as well as its suitability for application in the context of recreational maritime safety.

3. The manuscript cites a significant number of references, but these are not sufficiently integrated to support the research logic and discussions. It is recommended that the authors integrate Table 2 more effectively into the main text. For example, clarify how each dimension (software, hardware, etc.) is justified based on the cited references. Use subheadings or summary paragraphs to guide readers in understanding the standard-setting process.

Authors’ Reply: Thanks for the valuable comment. In response to this comment, we have revised the relevant content (please refer to the revised manuscript, pages 21 to 23) to provide a detailed explanation of each dimension of the SHELL model, including Software, Hardware, Environment, and Liveware. The corresponding criteria for each dimension have been clearly described and systematically incorporated into the main body of the manuscript.

4. On page 11, the authors mentioned total of 9 participants were selected from different fields. For a highly heterogeneous topic, this number is relatively small. It is recommend that the authors explain why 9 participants are sufficient to reach a consensus and address potential biases (such as local vs. international applicability).

Authors’ Reply: Thanks for the valuable comment. This study adopted purposive sampling, prioritizing expertise and perspective diversity over sample size in line with MDM principles. To ensure balanced representation, the panel comprised three experts each from government, academia, and the private sector, reflecting both regulatory and user-side views. Several members also had experience with international maritime safety frameworks, enhancing the cross-regional relevance of the findings. (please refer to the revised manuscript, page 20)

After the first round of the Delphi survey, the results indicated high levels of agreement: the CV for all criteria was below 0.5, and 95% of the criteria had a CV below 0.3. These results suggest a strong level of consensus (please refer to the revised manuscript, page 25). However, had there been significant divergence in expert opinions, a second round of the Delphi process would have been conducted to reach a stable consensus, in line with the fundamental principles of expert consultation in multi-criteria evaluation settings.

5. Pages 19, 20. The author shows that Figures 2 and 3 (INRM & Criteria Relationship Map) are important but not clearly explained. It is suggested that the author could add brief figure captions explaining the quadrant's meaning and how practitioners can interpret them. For example: "Top-right quadrant indicates criteria with high centrality and positive influence…"

Authors’ Reply: Thanks for the valuable comment. We have revised the figure captions to include explanations of the meaning represented by each of the four quadrants in the Influence Network Relation Map (INRM) and the Criteria Relationship Map, in order to assist readers in interpreting the analytical results (please refer to the revised manuscript, pages 32 to 34).

6. For the journal market and its international readership, discussions and case studies of this manuscript are limited to Taiwan and may restrict broader impact. Authors are encouraged to briefly discuss how this framework could be adapted to other countries facing similar regulatory gaps, such as those in Southeast Asia or the Pacific Islands.

Authors’ Reply: Thanks for the valuable comment. We have added a supplementary paragraph in the discussion section to briefly outline the potential applicability of the proposed analytical framework to other regions, such as Southeast Asia and the Pacific Islands (please refer to the revised manuscript, pages 42 to 45). In the conclusion section, we further summarize the contextual adaptability of the SHELL model when integrated with expert-based Delphi and DEMATEL methods. This integrated approach demonstrates its effectiveness in identifying systemic risks under conditions of weak regulatory capacity and offers a practical reference for policy planning, risk prioritization, and institutional development (please refer to the revised manuscript, page 47). These additions enhance the international and academic relevance of the study.

7. On pages 25-26 the conclusion part. The conclusion repeats the research findings but fails to highlight new content and its significant contributions. It is recommended that the authors add a summary of the findings, such as the innovative integration of SHELL and DEMATEL, a practical tool for policymakers in leisure marine safety, validation of recommendations using empirical events, etc.

Authors’ Reply: Thanks for the valuable comment. In response to this comment, we have revised the Conclusion section (please refer to the revised manuscript, pages 46 to 47) to emphasize the methodological contribution of integrating the SHELL model with the Delphi method and DEMATEL analysis. This combination is presented as a structured and practical tool for analyzing safety risks in the field of marine recreation. To strengthen the significance and applicability of our findings, we have also included reference to the previously mentioned 2024 incident and the implementation of a personnel safety monitoring system related to aquatic recreational activities, thereby reinforcing the study's practical relevance and potential impact.

8. Before considering accepting the article, it is recommended that some revisions be made.

Authors’ Reply: Thanks for the valuable comment. We sincerely thank the reviewer for the positive overall evaluation and valuable suggestions regarding this study. In response to the comments provided, we have carefully addressed and revised each point accordingly. We believe these revisions have significantly enhanced the quality and academic value of the manuscript, and we respectfully submit the revised version for your further consideration.

---

## [Decision Letter · Decision Letter 1]

21 Jul 2025

PONE-D-25-20340R1A Multi-Criteria Decision-Making Framework for Managing the Safety of Marine Recreational Powered Platforms: Integration with the SHELL ModelPLOS ONE

Dear Dr. Hsu,

Thank you for submitting your manuscript to PLOS ONE. After careful consideration, we feel that it has merit but does not fully meet PLOS ONE’s publication criteria as it currently stands. Therefore, we invite you to submit a revised version of the manuscript that addresses the points raised during the review process.

We look forward to receiving your revised manuscript.

Kind regards,

Yi-Che Shih, Ph.D.

Academic Editor

PLOS ONE

Journal Requirements:

**Additional Editor Comments:**

The revised manuscript is well-executed, methodologically sound, and directly relevant to safety governance in maritime recreation. With minor improvements in abstract clarity, table presentation, and brief global contextualization, the paper should be suitable for acceptance.

To further strengthen the manuscript before final acceptance:

• While improved, the abstract could be further enhanced by highlighting the unique contribution of the SHELL + DEMATEL integration in a single sentence near the end. Currently, it reads more like a summary than a pitch.

• Although a section is dedicated to discussing applicability to Southeast Asia and the Pacific Islands, the analysis is brief. Consider adding a short comparative note (1–2 sentences) on how regulatory gaps in one or two specific countries mirror Taiwan’s situation, with references if possible.

• Tables 4 and 7 are informative, but they are densely formatted. Adding clearer headings or footnotes for “centrality” and “net effect” values could improve reader comprehension.

Reviewers' comments:

Reviewer's Responses to Questions

**Comments to the Author**

1. If the authors have adequately addressed your comments raised in a previous round of review and you feel that this manuscript is now acceptable for publication, you may indicate that here to bypass the “Comments to the Author” section, enter your conflict of interest statement in the “Confidential to Editor” section, and submit your "Accept" recommendation.

Reviewer #3: All comments have been addressed

2. Is the manuscript technically sound, and do the data support the conclusions?

Reviewer #3: Yes

3. Has the statistical analysis been performed appropriately and rigorously? 

Reviewer #3: Yes

4. Have the authors made all data underlying the findings in their manuscript fully available?

Reviewer #3: Yes

5. Is the manuscript presented in an intelligible fashion and written in standard English?

Reviewer #3: Yes

6. Review Comments to the Author

Reviewer #3: This study addresses a significant research gap in the field of maritime safety management related to marine recreational activities. By integrating the SHELL model with DEMATEL in the maritime safety domain and utilizing the four-dimensional framework of the SHELL model (“Software,” “Hardware,” “Environment,” and “Liveware”), the research demonstrates interdisciplinary rigor and thus possesses innovative value. In the current revised version, all prior review comments have been adequately addressed, with each point systematically supplemented, explained, and discussed. Furthermore, substantial revisions have been made to both the methodological and theoretical aspects, which have effectively enhanced the validity and impact of this study.

7. PLOS authors have the option to publish the peer review history of their article (what does this mean? ). If published, this will include your full peer review and any attached files.

**Do you want your identity to be public for this peer review?** For information about this choice, including consent withdrawal, please see our Privacy Policy .

Reviewer #3: No

---

## [Author Response · Author response to Decision Letter 2]

3 Aug 2025

Additional Editor Comments Reply

Comment #1

• While improved, the abstract could be further enhanced by highlighting the unique contribution of the SHELL + DEMATEL integration in a single sentence near the end. Currently, it reads more like a summary than a pitch.

Authors’ Reply: Thanks for the valuable comment. In response to this suggestion, we have added the following sentence to the abstract section of the manuscript to highlight the unique contribution and methodological innovation of this study�"Building on the above findings, this study confirms the effectiveness of an innovative integration of the SHELL model and the DEMATEL method, which provides a structured and adaptive framework capable of systematically identifying systemic navigational risks in marine recreational activities."

(Please refer to the revised manuscript, pages ii, iii.)

Comment #2

• Although a section is dedicated to discussing applicability to Southeast Asia and the Pacific Islands, the analysis is brief. Consider adding a short comparative note (1–2 sentences) on how regulatory gaps in one or two specific countries mirror Taiwan’s situation, with references if possible.

Authors’ Reply: Thanks for the valuable comment. In addition to discussing the applicability of the research framework to Southeast Asia and Pacific Island nations, we further highlighted the similarities and regulatory gaps between these countries and Taiwan (Please refer to the revised manuscript, pages 41, 42). To enhance the completeness of the discussion, six additional references were incorporated (Please refer to the revised manuscript, page 51).

Comment #3

• Tables 4 and 7 are informative, but they are densely formatted. Adding clearer headings or footnotes for “centrality” and “net effect” values could improve reader comprehension.

Authors’ Reply: Thanks for the comment. We have revised the titles of Tables 4 and 7 to enhance clarity (Please refer to the revised manuscript, pages 24, 28) and added a footnote to Table 4 (Please refer to the revised manuscript, page 24) to improve reader comprehension.

---

## [Editor Report · Decision Letter 2]

6 Aug 2025

A Multi-Criteria Decision-Making Framework for Managing the Safety of Marine Recreational Powered Platforms: Integration with the SHELL Model

PONE-D-25-20340R2

Dear Dr. Hsu,

We’re pleased to inform you that your manuscript has been judged scientifically suitable for publication and will be formally accepted for publication once it meets all outstanding technical requirements.

Kind regards,

Yi-Che Shih, Ph.D.

Academic Editor

PLOS ONE

Additional Editor Comments (optional):

Congratution!
---

## [Editor Report · Acceptance letter]

PONE-D-25-20340R2

PLOS ONE

Dear Dr. Hsu,

I'm pleased to inform you that your manuscript has been deemed suitable for publication in PLOS ONE. Congratulations! Your manuscript is now being handed over to our production team.

Kind regards,

on behalf of

Dr. Yi-Che Shih

Academic Editor

PLOS ONE